# Evaluation of Cellulosic Polymers and Curcumin to Reduce Aflatoxin B1 Toxic Effects on Performance, Biochemical, and Immunological Parameters of Broiler Chickens

**DOI:** 10.3390/toxins11020121

**Published:** 2019-02-16

**Authors:** Bruno Solis-Cruz, Daniel Hernandez-Patlan, Victor M. Petrone, Karine P. Pontin, Juan D. Latorre, Eric Beyssac, Xochitl Hernandez-Velasco, Ruben Merino-Guzman, Casey Owens, Billy M. Hargis, Raquel Lopez-Arellano, Guillermo Tellez-Isaias

**Affiliations:** 1Laboratorio 5: LEDEFAR, Unidad de Investigación Multidisciplinaria, Facultad de Estudios Superiores Cuautitlán, Universidad Nacional Autónoma de México, Cuautitlan Izcalli 54714, Estado de Mexico, Mexico; bruno_sc@comunidad.unam.mx (B.S.-C.); danielpatlan@comunidad.unam.mx (D.H.-P.); vmpetrone@hotmail.com (V.M.P.); lopezar@unam.mx (R.L.-A.); 2Departamento de Medicina Veterinária Preventiva, Centro de Diagnóstico e Pesquisa em Patologia Aviária, Universidade Federal do Rio Grande do Sul, Porto Alegre RS 97105-900, Brazil; pontin.karine@gmail.com; 3Department of Poultry Science, University of Arkansas, Fayetteville, AR 72704, USA; juandlatorre@gmail.com (J.D.L.); cmowens@uark.edu (C.O.); bhargis@uark.edu (B.M.H.); 4Laboratoire de Biopharmacie et Technologie Pharmaceutique, UFR de Pharmacie, Faculté de Pharmacie, Université Clermont Auvergne, 63001 Clermont-Ferrand, France; eric.beyssac@uca.fr; 5Departamento de Medicina y Zootecnia de Aves, Facultad de Medicina Veterinaria y Zootecnia, Universidad Nacional Autónoma de Mexico, Mexico City 04510, Mexico; xochitl_h@yahoo.com (X.H.-V.); onirem@unam.mx (R.M.-G.)

**Keywords:** aflatoxin B1, broiler chickens, cellulosic polymers, curcumin, performance

## Abstract

To evaluate the effect of cellulosic polymers (CEL) and curcumin (CUR) on aflatoxin B1 (AFB1) toxic effects on performance, and the biochemical and immunological parameters in broiler chickens, 150 one-day-old male broiler chicks were randomly allocated into five groups with three replicates of 10 chickens per pen: Negative Control (feed); AFB1 (feed + 2 ppm AFB1); CUR (feed + 2 ppm AFB1 + Curcumin 0.2%); CEL (feed + 2 ppm AFB1 + 0.3% Cellulosic polymers); and, CEL + CUR (feed + 2 ppm AFB1 + 0.3% Cellulose polymers + 0.2% Curcumin). Every week, body weight, body weight gain, feed intake, and feed conversion ratio were calculated. On day 21, liver, spleen, bursa of Fabricius, and intestine from five broilers per replicate per group were removed to obtain relative organ weight. Histopathological changes in liver, several biochemical biomarkers, antibody titers, and muscle and skin pigmentation were also recorded. Dietary addition of 0.3% CEL and 0.2% CUR separately significantly diminished some of the toxic effects resulting from AFB1 on performance parameters, relative organs weight, histopathology, immune response, and serum biochemical variables (*P* < 0.05); however, the combination of CUR and CEL showed a better-integrated approach for the management of poultry health problems that are related with the consumption of AFB1, since they have different mechanisms of action with different positive effects on the responses of broiler chickens.

## 1. Introduction

Poultry and mycotoxins have been closely linked for several decades, as they are one of the most sensitive animal species to the toxic effects of mycotoxins. In the poultry industry, the ingestion of aflatoxin B1 (AFB1) lead to the impairment of all the essential productive parameters [1,2] and decreased resistance to common infectious diseases due to the depression of the humoral and cellular immune responses [3,4], resulting in substantial annual economic losses [5]. Furthermore, there is a public concern for the possible contamination with toxic residues into poultry products, such as meat and eggs, resulting in a potential hazard to human health, since the AFB1 residues may pass to these products at deficient levels [6].

Nowadays, there is no single method to counteract the toxic effects of AFB1, but different strategies and technologies have been implemented to address the challenges that are related to AFB1 and its toxic effects [7,8]. Undoubtedly, preventing the contamination of feed and grain with AFB1 is the best and first attempt; however, since prevention is not always possible, chemical, physical, and biological methods to decontaminate or detoxify feed and grains have been gaining attention [9,10,11]. Among these, the use of adsorbents is the most widely studied method as an alternative to diminish the toxic effects of AFB1 in animals, due to its economic practicality and its aptness for nutritional insight [12]. Additionally, since AFB1 cause cell damage, free radical production, and lipid peroxidation, the use of antioxidants acting as superoxide anion scavengers could aid in the overall detoxification process in the liver, and hence may help in the alleviation of aflatoxicosis [13,14]. In this sense, for practicality and effectiveness reasons, the most recent dietary approach to prevent aflatoxicosis in poultry is the combined use of adsorbents and antioxidants [15].

Curcumin (CUR), a polyphenolic curcuminoid, is one of the active natural products that have been extensively studied. CUR has been shown to possess antioxidant, anti-inflammatory, antimutagenic, antimicrobial, and anticancer properties [16,17]. Some researchers have demonstrated that CUR can ameliorate AFB1 toxicity through the increased activity of antioxidant enzymes, neutralizing the free radicals, and inhibiting AFB1 biotransformation into its 8,9-epoxide [18,19]. However, despite the demonstrated positive effects of CUR, its beneficial effects remain limited by its low solubility and poor bioavailability [20,21], which has encouraged researchers to use several strategies to increase both the solubility and bioavailability of CUR [22,23].

Previous research that was conducted in our laboratory has demonstrated that cellulosic polymers have great potential to adsorb AFB1 in vitro [24], but in vivo studies are needed to confirm the effectiveness of these materials. Furthermore, we have focused on the improvement of the solubility and permeability of CUR through the formation of a stable dispersion with polyvinylpyrrolidone (PVP) [25], with the results suggesting that the increase in solubility of CUR could be involved in the improvement of its antimicrobial activity. Hence, the objective of this study was to evaluate the effectiveness of cellulosic polymers, a stable dispersion of CUR, and the combination of both in reducing the adverse effects of AFB1 on performance parameters, serum biochemical parameters, cellular and humoral immunity parameters, and pathological damages of the liver in broiler chickens.

## 2. Results

The results of the performance parameters of Negative Control (NC), Positive Control (AFB1), and treated groups with Curcumin (CUR), Cellulosic polymers (CEL), and the combination of both (CEL+CUR) are summarized in Table 1. Apparent differences in body weight (BW) of the AFB1 group and the other groups began to be observed from day 14 of the study. At day 21, the body weights of the three treated groups and NC were significantly higher than AFB1 (*P* < 0.05). Although feed intake (FI) during the 21 days of the experiment was statistically similar for all of the groups, there were differences in body weight gain (BWG) that began in the second week of the study. At the end of the experiment, the BWG value of AFB1 group was approximately 150 g lower than NC and it was around 100 g below the rest of the treated groups. Therefore, the same trend was observed in the Feed Conversion Ratio (FCR), whose values were significantly higher for the AFB1 group as compared to the other groups. The detriment in performance parameters that was caused by 2 ppm AFB1 in the diet, which may be due to the impaired general health and the inhibitory effects of AFB1 on protein synthesis and lipidic metabolism, was diminished by the addition of the three treatments when compared to the control groups (Table 1).

Relative organ weights from birds that were used in the study are presented in Table 2. In the present study, chickens that received AFB1 showed a significant increase in the relative weight of their livers, since liver is the most sensitive organ to the effects of this mycotoxin. Interestingly, no significant differences in the weight of livers from birds given the three treatments as compared to NC were observed (Table 2). The weight of the intestine significantly increased in the AFB1 group when compared with the CEL and CEL+CUR groups, but just numerically when compared to the NC and CUR groups, making the intestine an organ relatively resistant to the toxicity of AFB1. The relative weight of spleen increased significantly only in birds from the AFB1 group and those that were treated only with CEL, as compared to the NC and CEL+CUR groups, which did not show a significant difference between them. The spleen weight of the group that was treated with CUR remained statistically similar to both control groups, while the CEL+CUR group was the only one that had no significant difference in comparison with the NC group. Regarding the relative weight of bursa of Fabricius, only the group treated with CUR remained similar to NC, with these two groups being significantly different from AFB1. The CEL and CEL+CUR groups showed no difference with any of the control groups. These observations in these two lymphoid organs are common in birds that were exposed to AFB1, in which lymphoid cell depletion and degeneration of follicle associated epithelium could be associated. The Spleen/bursa weight ratio showed the highest values for the AFB1 group, which was statistically different from the rest of the groups, suggesting that an increase in systemic inflammation may occur. In fact, unpublished studies that were conducted in our laboratory indicate that AFB1 affects intestinal permeability and increases liver bacterial translocation. However, it was remarkable to observe that the spleen/bursa ratio of all the treated groups were significantly lower than the AFB1 group, and the groups CUR and CEL+CUR reached spleen/bursa ratios statistically similar to the NC group (Table 2).

The histological analysis of the liver revealed a degree of hepatic injury that was significantly more critical in the AFB1 group, as compared to the NC group (Table 3). As shown in Figure 1, extensive vacuolar degeneration was observed in AFB1 group. Additionally, a change in the disposition and proliferation of cells in bile ducts near the liver portal space or among the hepatocytes was observed, with the presence of focal necrosis, as well as inflammatory cell infiltrates (Figure 2). Using any of the three treatments, since they were able to maintain significantly lower lesion scores than those in the AFB1 GROUP, diminished Hepatocellular degeneration and lymphoid infiltration. However, the combination CEL+CUR was the only treatment that reduced heterophilic infiltration to a lesion score that was statistically similar to the NC group.

In Table 4, the effects of 2 ppm AFB1 on some serum parameters are presented. When compared to the AFB1 group, SOD activity was significantly higher in the NC group. The SOD activity of the treated groups was also higher than the AFB1 group, but lower than the NC group, showing significantly improved antioxidant status in terms of superoxide dismutase activity. No statistical difference was observed in serum levels of mucin in any of the groups. For the serum levels of citrulline, there was no statistical difference between both control groups and the CEL and CEL+CUR groups. Only the CEL treated group was significantly higher when compared with the AFB1. As markers of intestinal function and absorption, these indicators did not show any affection in intestinal function. Regarding CRP levels, there was also no statistical difference between the control groups, but the treated groups had significantly lower concentration as compared to the AFB1 group, which suggests that there are less damaged cells when treatments were used, since the main function of this protein is to bind to the damaged cells and initiate their elimination.

For immunological parameters, the antibody titer against Newcastle disease and intestinal IgA levels were measured. In both cases, the AFB1 group showed significantly lower levels with respect to the NC group. The three treated groups were found to have antibody titers that are statistically similar to the NC group, while the intestinal IgA levels showed statistical similarity only between NC group and the group that was treated with the combination CEL+CUR. For CUR and CEL groups, intestinal IgA levels were only numerically higher than AFB1 group. These results suggest that including these treatments in the diet of broilers preserve both the systemic and local humoral immune response, since intestinal IgA level, which plays an important role in humoral immune responses of intestinal mucosa, and antibody titers against Newcastle disease were similar to that of the control group without aflatoxin. This effect was also observed for the cellular immune response, as determined by the CBH response that was increased with the use of the treatments evaluated, especially with the combination CEL+CUR (Table 5). Results from the AFB1 group confirmed the suppressive effect of AFB1 on the immune system, since there was a significant decrease in CBH response at both, 12 and 124 h, after the PHA-M injection as compared to the NC group, evidencing again that both cellular and humoral immunity may be adversely affected after feeding with AFB1 to poultry. At 12 h after the injection, the CUR treated group was not different from the AFB1 group, while the CEL and CEL+CUR groups were only numerically higher. At 48 h postinjection, the three treatments were numerically higher than AFB1, but only the group that was treated with the combination CEL+CUR was significantly different from the AFB1 group.

The effects of the evaluated treatments against 2 ppm AFB1 on serum biochemical variables are shown in Table 6. Feeding AFB1 caused significant changes in almost all the serum biochemical parameters between both control groups, except for AST and BUN. For ALP, there was no significant difference either, although there is a relatively clear numeric difference of more than 60 U/L between the control groups and lower levels for CUR and CEL treated groups as compared to the AFB1 group. A significant decrease in the serum levels of albumin, total proteins, cholesterol, triglycerides, glucose, creatinine, inorganic phosphorus, magnesium, and iron was observed in the AFB1 group in comparison with NC. CUR and CEL treated groups, as well the group that treated with the combination CEL+CUR, helped in diminishing the adverse effects of AFB1 maintaining higher serum concentrations of albumin, total proteins, cholesterol, glucose, inorganic phosphorous, and iron, in comparison to the AFB1 group.

Nevertheless, none of these treatments achieved the maintenance of similar serum values of triglycerides, creatinine, or magnesium to those values of the NC group, since they were somewhat statistically similar to AFB1 group. According to the results, an increase in the serum levels of ALT, GGT, and uric acid was observed in the AFB1 group in comparison with the NC group. The results also suggest a positive effect of the three treatments on these parameters, since the increase in the activity of both enzymes and uric acid levels were controlled with at least one of the treatments. For ALT and uric acid, all of the treated groups remained statistically like the NC group. For GGT, the group that was treated with CUR was the only group that remained statistically similar to NC group, whereas the CEL+CUR group resulted in only being numerically lower than the AFB1 group and the CEL group did not change when compared to the AFB1 group. The increased or reduced serum levels of these biochemical parameters are indicative of the toxic effect of AFB1 on hepatic, and maybe, renal tissues, or the result of a damaged metabolism, owing to disturbances in the mobilization, regulation, and use of these biomolecules and metabolites. 

Finally, the effect of the CUR inclusion as part of the diet on the skin of the thigh muscle and footpad color profiles is shown in Table 7. Dietary supplementation with CUR and CEL+CUR produced a significantly increased mean yellowness (b*) value in both, the skin of thigh muscle and footpad, of broilers (*P* < 0.05), in comparison with the rest of the groups that did not receive CUR as part of the diet, suggesting that there may be a greater absorption of CUR due to the improvement of its solubility, resulting in a better intestinal absorption, and therefore, a higher level of yellowness. The b* values in the footpad were higher than those in the skin of thigh muscle. In the thigh muscle, there were significant differences in lightness (L*) and redness (a*) among some of the groups. On the other hand, there was no significant difference in L* or a* from footpad amongst any of the study groups.

## 3. Discussion

One of the most economic impacts of AFB1 in poultry is the decrease of performance parameters. The results of this study are in agreement with those that were reported previously [26,27]. In the present study, the administration of 2 ppm AFB1 in poultry diets significantly decreased BW and BWG and increased FCR at the end of the 21-day feeding period, which must be related to alterations in protein and energy utilization, probably as a consequence of a deterioration of the digestive and metabolic efficiency of the birds [28,29]. The addition of CUR (0.2%) or CEL (0.3%) to the diet containing AFB1 significantly alleviated its adverse effects on these performance parameters, since these treated groups had no difference with the NC group, indicating that CUR and CEL did not negatively affect its nutritional integrity. Nevertheless, the combined inclusion of CUR (0.2%) and CEL (0.3%) to the diet did not result in a further amelioration of the toxic effects of AFB1 when compared with groups that were treated with CUR or CEL alone. Although it has been shown that CUR supplementation improves BW, BWG, and only shows better FCR results at high doses (0.5, 1.0, and 1.5%) [30,31,32], our results confirm the same effects, even with a much lower dose of CUR (0.2%), which could be the result of the improvement of its solubility and bioavailability through the use of a stable dispersion with PVP.

The results show that contamination with 2 ppm of AFB1 in feed caused a significant increase in the relative weight of liver and spleen, and a decrease in the relative weight of bursa of Fabricius, while no significant difference was observed in relative intestine weight. These results are consistent with findings of several studies [33,34], in which the increase in the relative weight of liver is attributed to lipid deposition, producing characteristic enlarged, friable, and fatty liver [35]. Moreover, the results of the relative weight of the liver are supported by the histopathological findings, which are also in agreement with previous studies [36,37]. Hepatocellular degeneration lesions that were recorded in the liver of AFB1 birds were the result of vacuolar degeneration and severe fat deposition (Figure 1), which could be due to impaired lipid transport rather than increased lipid biosynthesis [38]. In this group, we also observed severe inflammatory cell infiltrates mainly that were composed of lymphocytes and heterophils as a mechanism to respond to degenerate vacuolated hepatocytes [39]. In this study, the results of the relative weight of liver and histopathological findings that were obtained from the treated groups are close to the NC group, showing that the addition of CEL and CUR (0.2% and 0.3%, respectively) to the diets could be an option to decrease the detrimental effects of AFB1, either alone or in combination, mainly through the adsorption properties that CEL has shown [24], but also through the hepatoprotection properties of CUR [40].

On the other hand, a decrease in the relative weight of bursa of Fabricius may be caused by necrosis or cell depletion of this lymphoid organ, since it has been shown that bursal follicles are reduced in size during aflatoxicosis as a result of the depletion of both cortical and medullary lymphocytes, as well as a cell cycle arrest in the bursal cells of broilers [41], which might also explain the increase in the relative weight of spleen as a compensatory mechanism to the damage that is caused in the bursa of Fabricius [42]. This effect was confirmed with the spleen/bursa ratio, whose values are indicative of the development and growth on these lymphoid organs. The higher spleen/bursa ratio in the AFB1 group may indicate atrophy of the bursa and the increased migration of the lymphocyte subpopulations to the spleen and their proliferation [43,44], and then it can be used as a field indicator of the immune status. The addition of CUR and CEL in combination appeared to be effective in reducing the relative spleen weight, while CUR was the only treatment that maintained the weight of the Bursa de Fabricius as statistically similar to the NC group, evidencing its immunomodulatory effect by reducing the severity of lesions in this organ. Interestingly, both CUR and CEL, alone or in combination, showed values of the spleen/bursa ratio that were similar to the NC group, signifying that these treatments could help in decreasing the toxic effects of AFB1 on the bursa of Fabricius. There was no difference between the relative weights of the intestine in the control groups, suggesting that it is a dynamic organ that can adapt to a chronic AFB1 exposure, as has been demonstrated [45]. This finding confirms the results of serum mucin and citrulline, two biomarkers of the intestinal barrier health [46,47], which were not significantly modified in the control or treated groups (Table 4). 

SOD serum activity was determined to investigate the effect of the treatments against the oxidative stress that was induced by AFB1, since this enzyme is a fundamental part of the antioxidant system. In Table 4, our data show that SOD activity was markedly decreased in the AFB1 group when compared with that of the NC group, suggesting that broiler chicks that were treated with 2 ppm AFB1 may face a down-regulation of SOD gene expression, which is consistent with previous studies [48]. When compared with the AFB1 group, the SOD activity was higher in chicks that were fed the diet supplemented with the three treatments, although they could not reach SOD levels that were similar to those of the NC group, these data suggest that both CUR or CEL alone, but more efficiently their combination, counteract the oxidative damage that is caused by AFB1 through the adsorption of this mycotoxin by CEL and the anti-oxidant capacity of CUR to reduce lipid peroxidation and cell membrane damage [49].

The results of this study regarding the humoral and mucosal immunity agree with previous studies and confirm the immunosuppressive effect of AFB1 in broiler chickens [3,27]. AFB1 chickens showed the lowest antibody response to the ND vaccine and the total intestinal IgA concentration. This effect could be due to the capacity of AFB1 to inhibit RNA polymerase, resulting in a decrease in protein synthesis in general, but particularly immunoglobulins; or, to increased lysosomal digestion of immunoglobulins, with severe depletion and degeneration of lymphocytes in the bursal follicles and impairment of cytokines formation by lymphocytes [50,51]. CUR (0.2%), CEL (0.3%), and combination of both treatments were able to ameliorate the immunosuppressive effects of AFB1 with respect to the antibody titer against ND; nevertheless, the CEL+CUR combination (0.5%) was the only treatment that could maintain an intestinal IgA level that is comparable to the level of the NC group.

Likewise, it was clear from the results of the CBH test that feeding the birds with 2 ppm AFB1 implies a significant decrease in cellular immunity. The PHA skin test is used to evaluate cellular immunity, since PHA is a potent stimulant that activates immune cells infiltrating tissue from peripheral blood, causing local temporary inflammation at the injection site [52]. In the present study, AFB1 chickens also showed the lowest response to CBH at both, 12 and 24 h, after PHA injection. Interestingly, at 12 h after injection, no treatment was significantly different from the AFB1 group. However, by 24 h only, the CEL+CUR group had a CBH response that was statistically similar to that of the NC group, which indicates that neither CUR nor CEL alone could improve cellular immune function well, but a combination of both achieved an improvement in this response. These findings suggest that the immunomodulatory effects of CUR are mainly related to an improvement in humoral immune response, exerting either null or beneficial effects in cellular immune functions, due to its anti-inflammatory activity, whereby lymphocyte proliferation is inhibited and T cell-mediated immune functions are suppressed [53,54]. However, a combination of CUR with the adsorptive properties of CEL might also assist to ameliorate the AFB1 adverse effects in cellular immune response.

Regarding biochemical serum parameters, the results of total proteins and albumin were significantly diminished in the AFB1 group with respect to the NC group, which is an indicator of hepatic injury [55]. Since the liver mainly synthesizes serum proteins, including albumin, it is expected that hepatic injury that is caused by AFB1 consumption results in compromised protein synthesis. There are three probable mechanisms by which AFB1 might cause a decrease in the serum protein levels, which are related to the formation of adducts. AFB1 can form DNA or RNA-adducts that disturb transcription and translation in gene expression, and it can also form lysine adducts, resulting in proteins degradation or excretion, or by impairing messenger RNA synthesis by the selective inhibition of RNA polymerase II [56,57]. Even though CUR and CEL alone could ameliorate AFB1 toxic effects on serum levels of total proteins and albumin, it was the combination CEL+CUR that showed results that were statistically similar to the NC group. Changes in the serum enzyme activities of poultry resulting from hepatocellular injury that are caused by AFB1 have been previously reported [58]. Herein, chickens from AFB1 group receiving 2 ppm AFB1 showed a significant increase in serum activity of ALT and GGT enzymes, and although it was only numerical, there was also a clear rise in ALP activity. This increase in the levels of serum enzymes might be interpreted as a consequence of hepatocyte degeneration and subsequent leakage of enzymes into the bloodstream, as well as biliary cholestasis and the hyperplasia of bile ducts [55,56]. 

On the other hand, no significant changes were found in AST activity when birds where fed 2 ppm AFB1, which could be expected to occur, since AST is not exclusively a hepatic enzyme in domestic species, but it is also present in the cytoplasm and mitochondria of tissues, such as skeletal and cardiac muscles of all [59]. The serum enzyme activities of ALP and ALT were diminished by any of the evaluated treatments, either CUR or CEL alone, or the combination of both. Nevertheless, to reach a serum activity of GGT that was similar to the NC group, only CUR alone or in combination with CEL (CEL+CUR) worked, which once again shows the hepatoprotective activity of CUR, and the good adsorption properties of CEL against AFB1. The results show no significant difference in the BUN levels between AFB1 and NC groups when broilers were fed with 2 ppm AFB1, while there was a significant decrease in cholesterol, triglycerides, glucose, and creatinine serum levels, as well as an increase of uric acid. AFB1 has been reported to cause alterations in lipid metabolism, decreasing serum cholesterol and triglyceride levels, and our results coincide with those that have been previously reported [60,61]. A significant decrease in cholesterol and triglyceride serum levels might be attributed to the inhibition of cholesterol and fatty acid biosynthesis, concomitant with an inhibition of mobilization and transport of these lipids to peripheral tissue, resulting in an accumulation of these lipids in the liver [62,63], as described above. More recent studies have shown that AFB1 can downregulate the liver peroxisome proliferator activated receptor α (PPARα), a nuclear receptor protein that is a major regulator of lipid and glucose homeostasis, leading to an increase of the expression and activity of a lipolysis enzyme, lipoprotein lipase, and therefore promotes the clearance of triglyceride-rich lipoproteins, as well as circulating triglyceride levels [64]. Changes that were observed in cholesterol and triglyceride serum levels and the fatty liver observed in this study corroborate the deleterious effect of AFB1 in the lipid metabolism of broilers. Interestingly, neither CUR nor CEL alone could completely counteract these effects, even though numerically higher levels of these lipids were found in groups that were treated in comparison with the AFB1 group. Only the group that was treated with the combination of CEL+CUR reached cholesterol levels statistically similar to NC group, while none of the three treatments achieved triglyceride levels that were similar to those of the NC group. Both glucose and creatinine serum levels were decreased in AFB1 group when compared with the NC group, which is coincident with what has been previously reported when the birds were fed with AFB1 [64,65]. Decreased serum glucose might be a consequence of the reduced activity of enzymes that are involved in glucose metabolism, joined in the downregulation of PPARα by AFB1 [64,66]. On the other hand, low serum creatinine level results from the lower muscle mass gain, which is associated with severe hepatic injury during aflatoxicosis, since creatinine is a metabolite that results from the degradation of muscle phosphocreatine, so its level increases when muscle activity rates rise [64,67]. 

No treatment was able to maintain glucose serum levels that are similar to the NC group, there was an improvement with CUR and CEL alone, but the greatest increase in glucose serum levels was observed with the combination CEL+CUR. Nevertheless, the inclusion of these treatments was not enough to improve serum creatinine levels. An increase in the serum uric acid level in chicks from AFB1 group was observed with respect to those of the NC group, which might be an indication of impaired renal excretory function due to the toxic effect of AFB1 on the renal tissue [68,69]. Although no changes in creatinine serum levels were observed during 21 days of study, these would probably be increased in later days, since it has been reported that most pronounced changes in creatinine serum levels happen after 42 days of AFB1 exposure [70]. The results suggest that CUR and CEL supplementation counteracted the increase in serum uric acid levels, either individually or in combination. Serum mineral levels, such as P, Mg, and Fe, were significantly reduced after the birds consumed 2 ppm of AFB1, which is coincident with previous reports [61,71]. Lower levels of P and Mg may be due to the alteration of their metabolism, since AFB1 directly alters the renal, intestine, and parathyroid regulation of these minerals [72,73]. Moreover, it has also been reported that AFB1 produces changes in the serum Fe level, which is probably related to inflammation of the liver and perturbed proteins metabolism [74]. Nevertheless, the exact causes of the alteration in the mineral metabolism that is caused by AFB1 in broiler chickens have not been established and they cannot be elucidated from the results of this study. No treatment was completely capable of reversing the decreased serum mineral levels in the birds; however, both CUR and CEL alone or in combination were able to numerically improve the serum levels of P and Fe, while none of these helped to regulate the levels of Mg. According with the results of this study, improvement in most of the biochemical parameters of broilers that were fed with 2 ppm AFB1 corroborated the intrinsic capacity of CUR and CEL to maintain biochemical homeostasis [68,75,76,77], as well as their protective effects against AFB1, which might be due to the hepatoprotective and antioxidant activity of CUR and the adsorption ability of CEL in the digestive tract.

Lastly, the effect of supplementation with CUR on skin pigmentation was evaluated and then compared with NC group, since it has been previously reported that CUR has, in addition to the beneficial effects to counteract the toxicity of AFB1, positive effects on skin pigmentation of broiler chickens, which is an essential quality of poultry meat for consumers in many countries [78]. In the present study, we did not have much interest in skin redness, since it is not a good parameter when the birds are alive because the redness from blood vessels interferes with the reading. Moreover, the yellow pigments of CUR might have masked redness from the blood vessels and muscle, which is maybe the reason that NC group had the highest skin a* values [79]. Nonetheless, the results of this study showed that groups, including CUR in the diet as part of the treatment against toxic effects of AFB1, resulted in better skin pigmentation than those that did not receive it, producing a yellow color in the skin of thigh muscle and footpad of birds from these groups. Yellow color resulted to be higher in the skin of the footpad than the thigh muscle. This result may provide extra benefit to the use of CUR to achieve the aflatoxicosis control in poultry and increasing the skin color acceptance by the consumer.

The use of adsorbent materials against AFB1 is a field that has been explored in the last few decades; although most of these studies have been focused on inorganic materials, new organic adsorbents, or biopolymers, and synthetic polymers are being studied and recently tested [80]. Our laboratory has already demonstrated the adsorption efficiency of cellulosic polymers in vitro, with results suggesting that they have high adsorption capability for binding AFB1 [24]. The results of the in vivo study reported herein confirm the efficacy of cellulosic polymers to adsorb AFB1, and therefore to prevent its toxic effects in broiler chicks. Studies have shown that the main forces that are responsible for CEL adsorption capacity is a combination of electrostatic attractions and hydrogen bonding [81,82]. Since AFB1 is a very polar molecule, it is possible that electrostatic interactions between this mycotoxin and CEL are responsible for the formation of a mycotoxin monolayer on its surface, and therefore it is effective in preventing the deleterious effects of AFB1.

Moreover, an increased cellular immune response was observed with the addition of CEL into the diet, which coincides with previous reports of CEL immunomodulatory activity. This may result from the changes in gene expression of several molecules upon stimulation with CEL; molecules, such as nuclear factor kappa B (NF-κB), are activated via signaling pathways when CEL bind Toll-like and c-type lectin receptors, besides several genes within these pathways, are downregulated in their expression, which modulates cytokine profiles and immune status [83,84]. Additionally, as discussed above, CUR has been shown to be an effective natural chemoprotector against the toxic effects of AFB1 through three primary mechanisms, its antioxidant effects, immunomodulatory actions, and inhibitory activities against cytochrome P450 isoenzymes, without forgetting its significant hepatoprotective properties [40,85], which, due to the increase in its bioavailability, and thus, its efficiency and functionality were potentialized.

## 4. Conclusions

Data that were obtained from this study showed that feed broilers with 2 ppm of AFB1 caused severe toxic effects on productive, biochemical, and immunological parameters, as well as severe hepatic damage, which was evident from the results of performance parameters, relative organs weights, histopathology findings, immune response, and serum biochemical variables. Even though there are many methods to counteract the toxic effects that are caused by contamination with AFB1, there are not perfect strategies to eliminate this problem. Hence, new research must be done combining effective strategies to overcome the limitations that each technique has, leading to better prevention or even the elimination of aflatoxicosis problems in the poultry industry, improving food security, public health problems, and economic benefits. In this study, the dietary addition of 0.3% CEL and 0.2% CUR separately significantly diminished some of the toxic effects that resulted from AFB1. In the present study, the combination of CUR and CEL showed a better-integrated approach for the management of poultry health problems that are related with the consumption of AFB1, since they have different mechanisms of action with different positive effects on responses in broiler chickens.

## 5. Materials and Methods

### 5.1. Animal Source, Diets, and Experimental Design

One hundred fifty one-day-old male broiler chicks (Cobb-Vantress, Arkansas, USA) were raised in floor pens for 21 days. The chicks were neck-tagged, individually weighted, and randomly allocated to one of five groups: Negative Control (feed); AFB1 (feed + 2 ppm AFB1); CUR (feed + 2 ppm AFB1 + Curcumin 0.2%); CEL (feed + 2 ppm AFB1 + 0.3% Cellulosic polymers); and, CEL + CUR (feed + 2 ppm AFB1 + 0.3% Cellulose polymers + 0.2% Curcumin), each group had three replicates of 10 chickens (*n* = 30/group). Non-medicated mash corn-soybean-based broiler starter diet was formulated to approximate the nutritional requirements of broiler chickens, as recommended by the National Research Council [86] and then adjusted to breeder’s recommendations [87] (Table 8). AFB1 was added to the diets and mixed thoroughly to reach the specified concentration. Once prepared, each treatment was added to the experimental diet and then mixed thoroughly to the specified concentration. Birds had *ad libitum* access to water and feed. This study was carried out in accordance with the recommendations of Institutional Animal Care and Use Committee (IACUC) at the University of Arkansas, Fayetteville. The protocol #15006 was approved by the IACUC at the University of Arkansas, Fayetteville on 28 May 2015 for this study. 

### 5.2. Aflatoxin Production

AFB1 was provided by Dr. George E. Rottinghaus, Veterinary Medical Diagnostic Laboratory, University of Missouri, Columbia, MO 65211. AFB1 was produced through the fermentation of rice, according to the methodology that was previously described [88], using *Aspergillus parasiticus* NRRL (Northern Regional Research Laboratory) 2999 from Agriculture Research Service (ARS) culture collection, United States Department of Agriculture, and the aflatoxin content was measured by spectrophotometric analysis. The aflatoxin within the rice powder consisted of 74.62% AFB1, 22.38% AFG1, 2.48% AFB2, and 0.49% AFG2, based on total aflatoxin in the rice powder. Diets containing AFB1 were analyzed and the presence of parent AF was confirmed by the high-performance liquid chromatography with fluorescence detection (HPLC-FLD) method by using a Romer Derivatization Unit (Romer Labs Inc., Washington, MO, USA).

### 5.3. Preparation of Treatments

The CUR treatment was a stable dispersion that was prepared by dissolving CUR in a solution containing PVP in a 1:9 ratio, followed by the solvent evaporation at 40 °C for 48 h, and subsequently sieving. For CEL treatment, Microcrystalline Cellulose (MCC, Avicel™, FMC, Philadelphia, PA, USA), and Sodium Carboxymethylcellulose (CMC, Aqualon™, Ashland, Columbus, OH, USA) were combined in a 1:9 ratio, and then they were granulated, dried, and sieved. The last treatment was a mixture of the CUR and CEL treatments. In all of the treatments, the particle size was homogenized using a No. 25 mesh sieve to obtain particles with an average size of 700 µm.

### 5.4. Performance Parameters

Replicates in the experiment were used as experimental units for growth performance parameters. The birds were individually weighed on a weekly basis and the feed intake per pen was recorded at weekly intervals for 21 days. Body weight (BW), body weight gain (BWG), feed intake (FI), and feed conversion ratio (FCR) were calculated every week.

### 5.5. Relative Organs Weight

On day 21, all of the birds were humanely euthanized by CO_2_ inhalation. Liver, spleen, bursa of Fabricius, and intestine from 15 birds (five broilers from each replicate) per group were removed, cleaned of adherent tissues, rinsed with 0.9% saline solution, and then preserved at 4 °C until being weighed. The weight of these organs was measured and expressed as relative organ weights: Relative weight = (Organ weight)/(Final body weight) × 100. Spleen/bursa weight ratio was also calculated.

### 5.6. Evaluation of Aflatoxin B1 on Histological Lesions in Liver Tissue

Hepatocellular degeneration, lymphoid, and heterophilic infiltration were evaluated in the liver of the experimental birds. For this, livers from 12 birds from each treatment group (four chickens from each replicate) were removed on day 21. Livers were cleaned of adherent tissues, rinsed with 0.9% saline solution, and then fixed in 10% neutralized buffered formaldehyde. Once fixed, a transversal section of the middle part of the left liver lobule was processed routinely, dehydrated in increasing alcohol concentrations, and embedded in paraffin. Tissue samples were sectioned at five μm thicknesses, stained with hematoxylin and eosin, and mounted with coverslips for histological analysis. Hepatocellular degeneration was scored, as follows. Score: 0 = absence of cellular swelling; 0.5 = vacuolar degeneration and/or mild fat deposition; 1 = vacuolar degeneration and/or mild to moderate fat deposition; 2 = vacuolar degeneration and/or moderate fat deposition; 3 = vacuolar degeneration and/or moderate to severe fat deposition; and, 4 = vacuolar degeneration and/or severe fat deposition. The score was obtained by evaluating five fields with a magnification of 20x per tissue cut. The lesion score was obtained when it covered 50% or more of the cut. The median, mode, and variance of the 30 scores were calculated (five fields for twelve tissue cuts). 

The quantification of inflammatory cells (both lymphocytes and heterophils) was obtained using an adapted methodology that was previously described [89]. In a general field from the upper left end of the tissue cut, with the 5× objective, an area of 3.4 mm^2^ was evaluated. The total number of perivascular areas and clusters of inflammatory cells of each tissue cut were counted. Those fields with less than four perivascular areas or inflammatory cell clusters were excluded. To quantify the number of layers in the perivascular area, the largest radius containing a significant number of perivascular layers next to the center of the polygon in the vein was used. In lymphoid clusters, inflammatory cells were quantified by counting the number of cell layers when considering a radius resulting from the largest diameter of the cluster. The number of clusters multiplied the number of layers, and the average was then obtained. The lesion score was assigned by counting the number of cell layers, from the center of the cell cluster or the space of the perivascular area, towards the perimeter of the cluster where the greatest number of cell layers were present. The lesion score was used, as follows. Score: 0 = 0 cell layers per cell cluster (absence of inflammatory infiltrate); 0.5 = 1–11 cell layers per cell cluster (mild inflammatory infiltrate); 1 = 12–24 cell layers per cell cluster (mild to moderate inflammatory infiltrate); 2 = 25–50 cell layers per cell cluster (moderate inflammatory infiltrate); 3 = 51–100 cell layers per cell cluster (moderate to severe inflammatory infiltrate); and, 4 = more than 100 cell layers per cell cluster (severe inflammatory infiltrate). The median, mode, and variance of the total of the 30 scores were calculated (five fields for twelve tissue cuts). 

### 5.7. Serum Biochemical Analysis

Following the euthanize of birds on day 21, 12 birds from each treatment were randomly selected (four broilers from each replicate) and bled from the femoral vein before necropsy. Blood was centrifuged at 2500 rpm at 4 °C for 15 min; serum was separated and preserved at −20 °C until submitted for biochemical analysis. Serum concentrations of albumin, total protein, alkaline phosphatase (ALP), alanine aminotransferase (ALT), aspartate aminotransferase (AST), gamma-glutamyltransferase (GGT), blood urea nitrogen (BUN), cholesterol, triglycerides, glucose, creatinine, uric acid, phosphorus, magnesium, and iron were determined using a Corning clinical chemistry analyzer (Chiron Corporation, San Jose, CA, USA). Superoxide dismutase (SOD) activity, as well as mucin, citrulline, and C reactive protein (CRP) levels, were determined in serum samples using commercial assay kits and following the manufacturer's instructions. For SOD an assay kit was used to determine three types of SOD (Cu/Zn, Mn, and FeSOD) (Cayman chemical company, Ann Arbor, MI, USA. Catalog No. 706002), with an optimal dilution of the samples of 1:5. ELISA kits employing double antibody sandwich technique were used for the quantitative determination of mucin, citrulline, and CRP serum concentrations (MyBioSource, San Diego, CA, USA. Catalog No. MBS2505849, MBS2601045, and MBS261842, respectively), using undiluted serum samples for mucin and citrulline, and an optimal dilution of 1:20 for CRP. All of the samples were measured at 450 nm using an ELISA plate reader (Synergy HT, multi-mode microplate reader, BioTek Instruments, Inc., Winooski, VT, USA).

### 5.8. Intestinal IgA Levels

Intestinal IgA levels were determined in gut rinse samples, as previously described [90]. For this, intestine from 12 birds (four broilers from each replicate) per group was removed after their euthanize on day 21. Subsequently, 5 cm sections from Meckel's diverticulum were taken, they were rinsed, and then extruded three times with 5 mL 0.9% saline solution to obtain the intestinal mucosa. The rinse was collected in a tube and centrifuged at 3000 rpm at 4 °C for 10 min. The supernatant was retrieved and frozen at −20 °C until used. An indirect ELISA was performed to quantify IgA. The commercial chicken IgA ELISA quantitation set (Bethyl Laboratories Inc., Montgomery, TX, USA. Catalog No. E30-103) was used, according to the manufacturer's instructions. 96-well plates (Nunc MaxiSorp, Thermo Fisher Scientific, Rochester, NY, USA. Catalog No. 439454) were used. Samples were measured at 450 nm using an ELISA plate reader (Synergy HT, multi-mode microplate reader, BioTek Instruments, Inc., Winooski, VT, USA). The chicken IgA concentration that was obtained was multiplied by the dilution factor to determine the amount of chicken IgA in the undiluted samples. The optimum dilution for total IgA quantitation in gut rinse samples was 1:100.

### 5.9. Evaluation of Humoral Immunity: Antibody Production against Newcastle Disease Virus

On the first day of age, before all of the chicks were separated to form the groups, blood samples were taken, and the maternal antibody titers against Newcastle disease virus were determined using a commercial ELISA kit (AffiniTech, LTD., Bentonville, AR, USA. Catalog No. NDV-0200). Afterwards, all of the chicks were vaccinated with Newcastle Disease (ND)—Infectious Bronchitis (IB) vaccine (B1 ND strain plus Mass & Ark IB serotypes, Live Virus CEO, Merial, Athens, GA, USA) by ocular administration. On day 14, the broilers were vaccinated ocularly again with the same vaccine. On day 21, 12 birds (four broilers from each replicate) per treatment were randomly selected, humanely euthanized by CO_2_ inhalation, and bled from the femoral vein before necropsy. Blood was centrifuged at 2500 rpm at 4 °C for 15 min, and serum was separated and preserved at −20 °C until used. The antibody titers against ND were determined while using the same commercial ELISA kit.

### 5.10. Evaluation of Cellular Immunity: Skin Response to Phytohemagglutinin (PHA)

To evaluate the cellular immune activity, the phytohemagglutinin-induced cutaneous basophil hypersensitivity (CBH) response in the interdigital skin of birds was used. On day 18, 12 birds (four broilers per replicate) from each treatment were randomly selected and intradermally injected in the interdigital skin between the third and fourth digits of the right foot with 0.1 mL of PHA-M (Gibco, Grand Island, NY, USA. Catalog No. 10576015). The CBH response was evaluated by determining the thickness of the interdigital skin at the injection site with a digital caliper before injection and 12 and 24 h after the injection. The CBH response was calculated by:CBH response (mm) = (thickness 12 and 24 h postinjection, right foot) − (thickness preinjection, right foot)

### 5.11. Skin Pigmentation

The CIE system color profile of lightness (L*), redness (a*), and yellowness (b*) was measured by a reflectance colorimeter (Minolta Chroma Meter CR-300, Minolta Italia S.p.A., Milan, Italy) to evaluate skin pigmentation. For this, the color measurements of 12 birds (four broilers per replicate) from each treatment were carried out on the skin of both thigh muscle and footpad on day 20. 

### 5.12. Statistical Analysis

Data from performance, relative organs weight, serum biochemical analysis, intestinal IgA levels, humoral, and cellular immunity evaluation, as well as skin pigmentation, were subjected to ANOVA as a complete randomized design using the General Linear Models procedure of SAS [91]. Data are expressed as mean ± standard error, and significant differences among the means were determined by using Duncan's multiple range test at *P* < 0.05. Data from lesion scores of liver histopathological analysis are expressed as median (mode; variance) and the differences among median values of the groups were analyzed with the Mann–Whitney U test, with a level of significance set at *P* < 0.05.

## Figures and Tables

**Figure 1 toxins-11-00121-f001:**
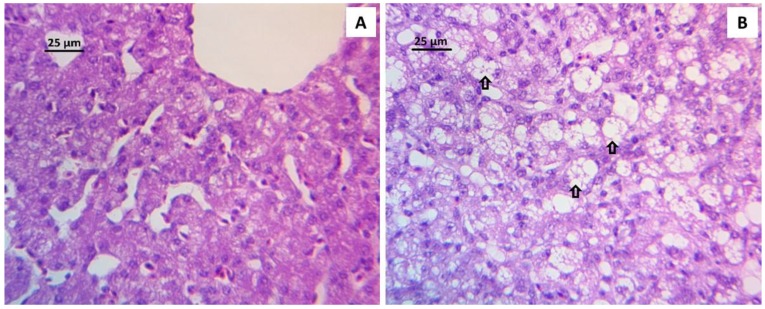
Liver histopathology showing the different values of the lesion score used to evaluate hepatocellular degeneration. (**A**) Hepatocellular degeneration score of 0.5: liver section from a bird of NC group showing a scarce number of intracytoplasmic vacuoles. (**B**) Hepatocellular degeneration score of 3.0: liver section from a bird of AFB1 group, after 21 days receiving a diet with 2 ppm of AFB1, showing an increase of the number of intracytoplasmic vacuoles (arrows). Stain: Hematoxylin and Eosin.

**Figure 2 toxins-11-00121-f002:**
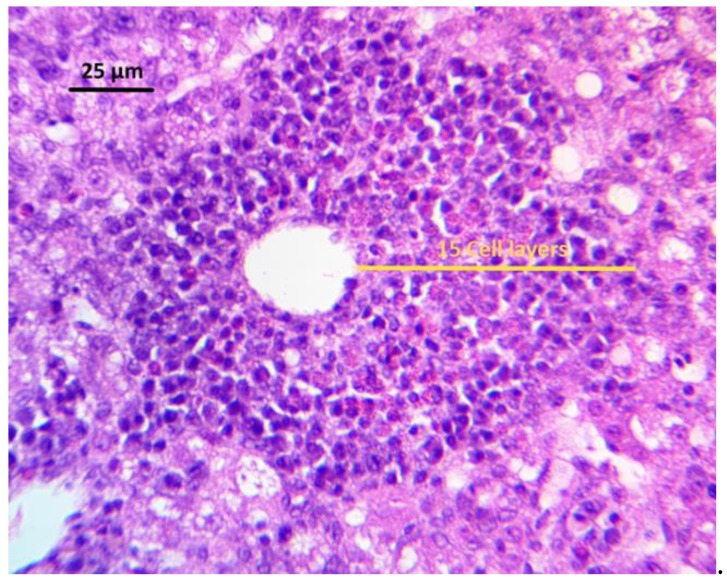
Inflammatory cell layers (heterophiles) around hepatic perivascular spaces in a liver section from a bird of AFB1 group, after 21 days receiving a diet with 2 ppm of AFB1. The yellow line indicates that the counted inflammatory cell layers contained in the radius with the greatest number of cell layers, when considering the center of the perivascular space as the origin of the radius.

**Table 1 toxins-11-00121-t001:** Evaluation of body weight (BW), body weight gain (BWG), feed intake (FI), and feed conversion ratio (FCR) in broiler chickens consuming a corn-soybean based diet contaminated with AFB1 (2 ppm) supplemented with the three treatments ^1^.

Item	NC	AFB1	CUR	CEL	CEL + CUR	SEM^2^	*P*-Value
**BW, g/broiler**							
day 0	46.48 ± 0.66^a^	47.77 ± 0.66^a^	48.12 ± 0.83^a^	48.46 ± 0.63^a^	46.71 ± 0.73^a^	0.3198	0.2181
day 7	135.28 ± 4.37^b^	132.28 ± 2.79^b^	138.78 ± 3.79^b^	154.56 ± 3.96^a^	132.07 ± 3.33^b^	1.7493	<0.0001
day 14	326.82 ± 17.18^a^	286.57 ± 10.30^b^	318.17 ± 13.33^ab^	349.09 ± 12.03^a^	325.24 ± 11.73^a^	5.9347	0.0165
day 21	651.28 ± 30.67^a^	500.48 ± 18.30^b^	590.47 ± 20.67^a^	615.06 ± 20.74^a^	604.57 ± 18.99^a^	10.5226	<0.0001
**BWG, g/broiler**							
day 0–7	88.80 ± 4.02^b^	84.52 ± 2.74^b^	90.66 ± 3.42^b^	106.09 ± 3.90^a^	85.36 ± 3.28^b^	1.6593	0.0001
day 7–14	191.54 ± 13.79^a^	154.28 ± 8.49^b^	179.40 ± 10.71^ab^	194.54 ± 9.45^a^	193.17 ± 9.45^a^	4.7468	0.0293
day 14–21	324.46 ± 15.81^a^	213.92 ± 10.01^c^	272.29 ± 12.27^b^	265.96 ± 13.59^b^	279.33 ± 10.09^b^	6.1797	<0.0001
day 0–21	604.80 ± 30.43^a^	452.72 ± 18.40^b^	542.34 ± 20.52^a^	566.59 ± 20.65^a^	557.86 ± 19.02^a^	10.5017	<0.0001
**FI, g/broiler**							
day 0–7	138.60 ± 4.03^b^	160.17 ± 3.16^a^	161.17 ± 4.36^a^	157.35 ± 7.11^a^	133.40 ± 4.90^b^	3.6393	0.0054
day 7–14	219.50 ± 8.81^a^	237.25 ± 13.63^a^	241.42 ± 3.44^a^	233.05 ± 6.75^a^	234.13 ± 8.34^a^	3.8787	0.5135
day 14–21	392.45 ± 14.54^a^	378.52 ± 11.89^a^	383.63 ± 11.36^a^	360.48 ± 28.15^a^	391.97 ± 7.95^a^	6.9276	0.6461
day 0–21	750.55 ± 17.23^a^	775.93 ± 3.51^a^	786.22 ± 17.42^a^	750.88 ± 39.17^a^	759.50 ± 21.15^a^	9.4079	0.7466
**FCR**							
day 0–7	1.08 ± 0.06^abc^	1.21 ± 0.03^a^	1.17 ± 0.05^ab^	1.05 ± 0.05^bc^	1.02 ± 0.01^c^	0.0250	0.0420
day 7–14	1.20 ± 0.07^b^	1.39 ± 0.05^a^	1.28 ± 0.03^ab^	1.18 ± 0.02^b^	1.15 ± 0.02^b^	0.0288	0.0215
day 14–21	1.27 ± 0.07^b^	1.55 ± 0.05^a^	1.35 ± 0.03^b^	1.29 ± 0.01^b^	1.27 ± 0.02^b^	0.0326	0.0036
day 0–21	1.38 ± 0.08^b^	1.72 ± 0.07^a^	1.47 ± 0.04^b^	1.41 ± 0.02^b^	1.38 ± 0.03^b^	0.0390	0.0037

^1^ Data are expressed as mean ± SE. ^2^ Standard error of the means. ^a–c^ Superscripts within rows indicate significant difference (*P* < 0.05), according Duncan’s multiple range tests; *n* = 30.

**Table 2 toxins-11-00121-t002:** Relative weight of liver, spleen, bursa of Fabricius and intestine in broiler chickens consuming AFB1 (2 ppm) contaminated feed for 21 days supplemented with the three treatments ^1^.

Relative Weight (g)	NC	AFB1	CUR	CEL	CEL + CUR	SEM ^2^	*P*-Value
Liver	3.245 ± 0.101^b^	4.200 ± 0.158^a^	3.558 ± 0.179^b^	3.418 ± 0.147^b^	3.282 ± 0.111^b^	0.0739	<0.0001
Intestine	9.635 ± 0.499^ab^	10.620 ± 0.362^a^	9.326 ± 0.529^ab^	8.813 ± 0.444^b^	8.873 ± 0.333^b^	0.2060	0.0330
Spleen	0.119 ± 0.008^b^	0.159 ± 0.011^a^	0.136 ± 0.008^ab^	0.157 ± 0.016^a^	0.123 ± 0.010^b^	0.0051	0.0257
Bursa of Fabricius	0.281 ± 0.021^a^	0.214 ± 0.010^b^	0.272 ± 0.016^a^	0.264 ± 0.017^ab^	0.252 ± 0.023^ab^	0.0083	0.0924
Spleen/bursa ratio	0.433 ± 0.020^c^	0.755 ± 0.050^a^	0.528 ± 0.050^bc^	0.600 ± 0.040^b^	0.527 ± 0.047^bc^	0.022	<0.0001

^1^ Data are expressed as mean ± SE. ^2^ Standard error of the means. ^a–c^ Superscripts within rows indicate significant difference (*P* < 0.05), according Duncan’s multiple range tests; *n* = 15.

**Table 3 toxins-11-00121-t003:** Hepatocellular degeneration and inflammatory cells infiltration in livers from broiler chickens consuming AFB1 (2 ppm) contaminated feed during 21 days supplemented with the three treatments ^1^.

Item	NC	AFB1	CUR	CEL	CEL + CUR
Hepatocellular degeneration	2.50 (2.00; 1.37)^a^	3.50 (3.00; 1.07)^b^	2.00 (2.00; 0.40)^a^	2.00 (2.00; 0.50)^a^	1.50 (1.00; 1.60)^a^
Lymphoid infiltration	1.00 (1.00; 0.40)^a^	3.50 (4.00; 1.37)^b^	1.50 (1.00; 0.67)^a^	2.00 (1.50; 0.70)^a^	2.00 (2.00; 0.27)^a^
Heterophilic infiltration	0.50 (1.00; 0.30)^a^	1.50 (1.00; 3.87)^b^	2.50 (2.50; 1.87)^b^	3.00 (3.00; 1.80)^b^	1.00 (1.00; 0.97)^ab^

^1^ Data are expressed as median (mode; variance). ^a,b^ Superscripts within rows indicate significant difference (*P* < 0.05), according Mann-Whitney U test; *n* = 12.

**Table 4 toxins-11-00121-t004:** Effect of the three treatments on serum levels of superoxide dismutase (SOD), mucin, citrulline and C-reactive protein (CRP), and the antibody titers against Newcastle disease (ND) and intestinal IgA levels in broiler chickens consuming a corn-soybean based diet contaminated with AFB1 (2 ppm) for 21 days ^1^.

Group	SOD (U/mL)	Mucin (pg/mL)	Citrulline (nmol/mL)	CRP (ng/mL)	ND Titer	Intestinal IgA (µg/mL)
NC	13.33 ± 0.15^a^	104.00 ± 28.25^a^	22.30 ± 3.54^ab^	25.10 ± 1.35^ab^	1327.4 ± 99.70^a^	42.47 ± 7.46^a^
AFB1	9.83 ± 0.81^c^	68.67 ±14.24^a^	13.90 ± 1.97^b^	30.18 ± 4.11^a^	846.5 ± 29.03^b^	25.47 ± 2.38^b^
CUR	11.32 ± 0.49^bc^	62.67 ± 7.06^a^	21.40 ± 3.19^ab^	21.86 ± 2.31^b^	1258.8 ± 112.72^a^	36.39 ± 2.84^ab^
CEL	11.50 ± 0.67^b^	92.00 ± 15.90^a^	26.27 ± 3.77^a^	22.51 ± 1.87^b^	1467.2 ± 126.66^a^	39.62 ± 5.28^ab^
CEL + CUR	11.45 ± 0.13^b^	73.67 ± 16.90^a^	17.72 ± 2.89^ab^	20.88 ± 2.13^b^	1313.9 ± 165.06^a^	41.43 ± 4.72^a^
SEM^2^	0.2753	7.9510	1.5035	1.1861	57.8831	2.2718
*P*-value	0.0010	0.4522	0.0793	0.0871	0.0063	0.1129

^1^ Each value represents the mean ± standard error. ^2^ Standard error of the means. ^a–c^ Mean values in the same column that do not share a common letter differ significantly (*P* < 0.05), according Duncan’s multiple range tests; *n* = 12.

**Table 5 toxins-11-00121-t005:** Cutaneous basophil hypersensitivity (CBH) response induced by PHA-M in broiler chickens consuming a corn-soybean based diet contaminated with AFB1 (2 ppm) supplemented with the three treatments ^1^.

Group	12 h (mm)	24 h (mm)
NC	0.692 ± 0.052^a^	0.788 ± 0.072^a^
AFB1	0.486 ± 0.041^b^	0.577 ± 0.057^b^
CUR	0.481 ± 0.060^b^	0.702 ± 0.043^ab^
CEL	0.555 ± 0.040^ab^	0.719 ± 0.036^ab^
CEL + CUR	0.617 ± 0.047^ab^	0.741 ± 0.050^a^
SEM^2^	0.023	0.025
*P*-value	0.0151	0.0798

^1^ Each value represents the mean ± standard error. ^2^ Standard error of the means. ^a,b^ Mean values in the same column that do not share a common letter differ significantly (*P* < 0.05), according Duncan’s multiple range tests; *n* = 12.

**Table 6 toxins-11-00121-t006:** Effect of the three treatments on serum biochemical parameters in broiler chickens consuming a corn-soybean based diet contaminated with AFB1 (2 ppm) for 21 days ^1^.

Item	NC	AFB1	CUR	CEL	CEL + CUR	SEM^2^	*P*-value
Albumin (g/dL)	1.12 ± 0.03^a^	0.63 ± 0.07^c^	0.82 ± 0.07^b^	0.71 ± 0.07^bc^	1.00 ± 0.03^a^	0.0360	<0.0001
Total proteins (g/dL)	2.09 ± 0.06^a^	1.61 ± 0.18^b^	2.27 ± 0.18^a^	1.96 ± 0.11^ab^	2.34 ± 0.10^a^	0.0694	0.0040
ALP ^3^ (U/L)	278.10 ± 19.96^ab^	344.50 ± 28.76^a^	247.40 ± 27.30^b^	242.40 ± 34.21^b^	299.90 ± 25.14^ab^	12.9300	0.0712
ALT ^4^ (U/L)	1.78 ± 0.22^b^	3.49 ± 0.59^a^	1.16 ± 0.24^b^	1.24 ± 0.24^b^	0.96 ± 0.23^b^	0.1952	<0.0001
AST ^5^ (U/L)	201.43 ± 7.48^ab^	200.39 ± 14.45^ab^	175.02 ± 7.49^b^	194.66 ± 10.17^ab^	207.43 ± 6.92^a^	4.4586	0.1797
GGT ^6^ (U/L)	12.70 ± 0.68^b^	15.30 ± 0.76^a^	13.10 ± 0.62^b^	15.20 ± 0.76^a^	13.90 ± 0.57^ab^	0.3295	0.0272
BUN ^7^ (mg/dL)	2.89 ± 0.07^a^	2.54 ± 0.19^ab^	2.54 ± 0.14^ab^	2.69 ± 0.20^ab^	2.29 ± 0.15^b^	0.0732	0.1170
Cholesterol (mg/dL)	115.20 ± 4.95^a^	66.70 ± 8.41^c^	104.50 ± 8.95^ab^	91.30 ± 11.09^b^	126.20 ± 3.58^a^	4.4809	<0.0001
Triglycerides (mg/dL)	135.10 ± 11.91^a^	69.50 ± 6.44^b^	70.50 ± 6.98^b^	80.90 ± 10.11^b^	96.20 ± 6.97^b^	5.1114	<0.0001
Glucose (mg/dL)	422.00 ± 18.09^a^	287.50 ± 12.60^d^	331.10 ± 9.69^bc^	297.20 ± 12.29^cd^	353.40 ± 12.69^b^	8.9404	<0.0001
Creatinine (mg/dL)	0.28 ± 0.01^a^	0.20 ± 0.00^b^	0.20 ± 0.01^b^	0.18 ± 0.02^b^	0.18 ± 0.02^b^	0.0090	0.0006
Uric acid (mg/dL)	11.56 ± 0.72^b^	13.61 ± 0.89^a^	9.66 ± 0.32^b^	11.29 ± 0.71^b^	11.12 ± 0.49^b^	0.3350	0.0032
P ^8^ (mg/dL)	9.04 ± 0.29^a^	7.36 ± 0.41^b^	7.88 ± 0.41^ab^	8.59 ± 0.51^a^	8.48 ± 0.30^ab^	0.1883	0.0381
Mg ^9^ (mEq/L)	3.75 ± 0.15^a^	2.72 ± 0.10^b^	2.57 ± 0.12^b^	2.99 ± 0.21^b^	2.98 ± 0.21^b^	0.0913	<0.0001
Fe ^10^ (µg/dL)	130.30 ± 5.21^a^	95.40 ± 8.94^b^	116.70 ± 5.95^ab^	110.70 ± 4.22^ab^	120.80 ± 10.09^a^	3.5183	0.0212

^1^ Data are expressed as mean ± SE. ^2^ Standard error of the means. ^3^ ALP: alkaline phosphatase; ^4^ ALT: alanine aminotransferase; ^5^ AST: aspartate aminotransferase; ^6^ GGT: gamma glutamyltransferase; ^7^ BUN: blood urea nitrogen; ^8^ P: phosphorus; ^9^ Mg: magnesium; ^10^ Fe: iron. ^a–d^ Superscripts within rows indicate significant difference (*P* < 0.05), according Duncan’s multiple range tests; *n* = 10.

**Table 7 toxins-11-00121-t007:** Effect of dietary supplementation with the three treatments on skin of thigh muscle and footpad color of broiler chickens consuming a corn-soybean based diet contaminated with AFB1 (2 ppm) ^1^.

Item	NC	AFB1	CUR	CEL	CEL + CUR	SEM^2^	*P*-Value
**Thigh muscle**							
L*	71.14 ± 0.47^ab^	72.27 ± 0.50^a^	70.55 ± 0.28^bc^	70.98 ± 0.38^b^	69.71 ± 0.38^c^	0.2118	0.0015
a*	3.36 ± 0.43^a^	2.15 ± 0.29^b^	1.89 ± 0.29^b^	1.88 ± 0.31^b^	1.79 ± 0.17^b^	0.1460	0.0011
b*	2.75 ± 0.56^c^	4.89 ± 0.41^b^	7.83 ± 0.57^a^	6.13 ± 0.29^b^	7.54 ± 0.29^a^	0.3270	<0.0001
**Footpad**							
L*	70.89 ± 0.76^a^	70.94 ± 0.53^a^	71.65 ± 0.97^a^	70.66 ± 0.58^a^	70.66 ± 0.32^a^	0.2978	0.5395
a*	2.36 ± 0.26^a^	2.03 ± 0.32^a^	1.92 ± 0.11^a^	2.56 ± 0.44^a^	1.82 ± 0.31^a^	0.1376	0.4001
b*	13.08 ± 0.59^b^	13.65 ± 0.34^b^	19.12 ± 0.96^a^	14.66 ± 0.55^b^	17.25 ± 1.36^a^	0.4875	<0.0001

^1^ Each value represents the mean ± standard error. ^2^ Standard error of the means. ^a–c^ Mean values in the same row that do not share a common letter differ significantly (*P* < 0.05), according Duncan’s multiple range tests. L* = lightness; a* = redness; b* = yellowness; *n* = 12.

**Table 8 toxins-11-00121-t008:** Ingredient composition and nutrient content of a basal starter diet used for the study.

Item	Corn Soybean-Based Diet
Ingredients (g/kg)	
Corn	574.5
Soybean meal	346.6
Poultry oil	34.5
Dicalcium phosphate	18.6
Calcium carbonate	9.9
Salt	3.8
DL-Methionine	3.3
L-Lysine HCl	3.1
Threonine	1.2
Choline chloride 60%	2.0
Vitamin premix^1^	1.0
Mineral premix^2^	1.0
Antioxidant^3^	0.5
**Calculated analysis**	
Metabolizable energy (MJ/kg)	12.7
Crude protein (g/kg)	221.5

^1^ Vitamin premix supplied per kg of diet: retinol, 6 mg; cholecalciferol, 150 µg; dl-α-tocopherol, 67.5 mg; menadione, 9 mg; thiamine, 3 mg; riboflavin, 12 mg; pantothenic acid, 18 mg; niacin, 60 mg; pyridoxine, 5 mg; folic acid, 2 mg; biotin, 0.3 mg; cyanocobalamin, 0.4 mg. ^2^ Mineral premix supplied per kg of diet: Mn, 120 mg; Zn, 100 mg; Fe, 120 mg; Cu, 10–15 mg; I, 0.7 mg; Se, 0.2 mg; and Co, 0.2 mg. ^3^ Ethoxyquin.

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
