# Peer review of "Evaluation of Cellulosic Polymers and Curcumin to Reduce Aflatoxin B1 Toxic Effects on Performance, Biochemical, and Immunological Parameters of Broiler Chickens"

_toxins, 2019, doi:10.3390/toxins11020121_

Round 1
Reviewer 1 Report
The authors have an interesting story to tell and I am intrigued by the results they show. I have the following comments which I think can improve the manuscript:
1) How do the authors think CEL and/or CUR is working? Can they provide a working model for this study to provide a plausible mechanism of action?
2) I would like to see some gene expression and/protein biomarrker studies that can support the histopathological observations and can shed some light on mechanism of action
3) could the authors please add some interpretation at the end of each result? That way readers can understand what each results suggest. For example when spleen weight increases with AFB1 it would help if the authors add an explanation eg. say that results suggest an increase in systemic inflammation. That will help readers outside the field tremendously.
I think this study should be published here. The comments above will help the readers. Very nice work on statistics!
Author Response
ANSWERS TO REVIEWER 1
We thank you very much for the time you have spent on reviewing our manuscript. We have given full consideration to your comments and the manuscript that has been carefully revised and modified accordingly. Please refer to the point-by-point reply to the Reviewer comments.
1) How do the authors think CEL and/or CUR is working? Can they provide a working model for this study to provide a plausible mechanism of action?
As stated in the paper, from the results obtained in our previous studies and this in vivo study, the hypothesis we established is that the high adsorption capability of CEL to bind AFB1 is due to a combination of electrostatic attractions and hydrogen bonding between CEL and AFB1, which is a very polar molecule. Thus, it is possible that these electrostatic interactions between this mycotoxin and CEL are responsible for the formation of a mycotoxin monolayer on its surface. Nevertheless, this assumption must be verified, which we are currently doing. We are working on the physicochemical characterization which can provide information to clarify and understand the adsorption mechanism.
On the other hand, CUR has been reported as a powerful hepatoprotector, antioxidant, and immunomodulator [1–6], three of the main causes of AFB1-induced toxicity. However, since CUR beneficial effects are not effective enough due to its poor solubility and, therefore, low bioavailability, we attribute the excellent effects observed in this study to the improvement in the solubility of the CUR through the use of the solid dispersion.
2) I would like to see some gene expression and/protein biomarker studies that can support the histopathological observations and can shed some light on the mechanism of action.
Unfortunately, we did not run any tests of gene expression and/or protein biomarkers; however, there are already published studies with this information [7–9], whose results agree with our measurements of the proteins already expressed by these genes. So, we can conclude that biochemical measurements can also support our histopathological observations. Nevertheless, we are definitely going to consider this type of testing for the studies that we are currently carrying out. Thank you.
3) Could the authors please add some interpretation at the end of each result? That way readers can understand what each result suggests. For example, when spleen weight increases with AFB1 it would help if the authors add an explanation eg. say that results suggest an increase in systemic inflammation. That will help readers outside the field tremendously.
Suggestion accepted. We have included a brief interpretation of what the results suggest at the end of each result, hence the results section has been modified accordingly. Thank you.
References
1. Antony S, Kuttan R, Kuttan G. Immunomodulatory activity of curcumin. Immunological investigations. Taylor \& Francis; 1999;28(5-6):291–303.
2. Reyes-Gordillo K, Shah R, Lakshman M, Flores-Beltrán R, Muriel P. Hepatoprotective properties of curcumin. Liver Pathophysiology. Elsevier; 2017. p. 687–704.
3. Samarghandian S, Azimi-Nezhad M, Farkhondeh T, Samini F. Anti-oxidative effects of curcumin on immobilization-induced oxidative stress in rat brain, liver and kidney. Biomedicine \& Pharmacotherapy. Elsevier; 2017;87:223–9.
4. Yadav V, Mishra K, Singh D, Mehrotra S, Singh V. Immunomodulatory effects of curcumin. Immunopharmacology and immunotoxicology. Taylor \& Francis; 2005;27(3):485–97.
5. Yilmaz S, Kaya E, Kisacam MA. The Effect on Oxidative Stress of Aflatoxin and Protective Effect of Lycopene on Aflatoxin Damage. Aflatoxin-Control, Analysis, Detection and Health Risks. InTech; 2017.
6. Aggarwal BB, Surh Y-J, Shishodia S. The molecular targets and therapeutic uses of curcumin in health and disease. Springer Science \& Business Media; 2007.
7. Yarru L, Settivari R, Gowda N, Antoniou E, Ledoux D, Rottinghaus G. Effects of turmeric (Curcuma longa) on the expression of hepatic genes associated with biotransformation, antioxidant, and immune systems in broiler chicks fed aflatoxin. Poultry Science. Oxford University Press Oxford, UK; 2009;88(12):2620–7.
8. Abdulbaqi NJ, Dheeb BI, Irshad R. Expression of Biotransformation and Antioxidant Genes in the Liver of Albino Mice after Exposure to Aflatoxin B1 and an Antioxidant Sourced from Turmeric (Curcuma longa). Jordan Journal of Biological Sciences. 2018;11(1):93–8.
9. Limaye A, Yu R-C, Chou C-C, Liu J-R, Cheng K-C. Protective and detoxifying effects conferred by dietary selenium and curcumin against AFB1-mediated toxicity in livestock: a review. Toxins. Multidisciplinary Digital Publishing Institute; 2018;10(1):25.

Reviewer 2 Report
The paper entitled: “Evaluation of cellulosic polymers and curcumin to reduce aflatoxin B1 toxic effects on performance, biochemical and immunological parameters of broiler chickens” investigate the effect of cellulosic polymers (CEL) and curcumin (CUR) in alleviating the toxic effects of aflatoxin B1 on performance, biochemical and immunological parameters in broiler chickens. The study is of a great interest as aflatoxin B1 is a frequent contaminant of cereals and other food and feed commodities and there is a need to identify the most appropriated methods for counteract its toxic effects in animals and human. The solution proposed by this paper is original consisting in the dietary addition of cellulose and curcumin but also of their combination as dietary solution for counteracting the AFB1 effect. The paper is well written in a good English and the results are very well presented and correctly interpreted in the discussion section.
Minor observations
M&M section
Please provide more details about fermentation, at least the strain of Aspergillus used for AFB1 production.
Was the control diets analyzed for the presence of other mycotoxins?
Author Response
ANSWERS TO REVIEWER 2
We thank you very much for the time you have spent on reviewing our manuscript. We have given full consideration to your comments and the manuscript that has been carefully revised and modified accordingly. Please refer to the point-by-point reply to the Reviewer comments.
Please provide more details about fermentation, at least the strain of Aspergillus used for AFB1 production.
Suggestion accepted. We have included in the material and method section the following information regarding the fermentation process: AFB1 was produced through the fermentation of rice, according to the methodology previously described [88], using Aspergillus parasiticus NRRL (Northern Regional Research Laboratory) 2999 from Agriculture Research Service (ARS) culture collection, United States Department of Agriculture, and the aflatoxin content was measured by spectrophotometric analysis. Thank you.
Was the control diet analyzed for the presence of other mycotoxins?
Unfortunately, we did not analyze the presence of other mycotoxins in the feed used to perform this study. Nevertheless, the raw materials used for its manufacture guarantee mycotoxins levels below the limits or tolerances established to protect from harmful effects of mycotoxins. Furthermore, the same feed was used for all groups, including both positive and negative control, maintaining this parameter as a constant during the experiment in all the experimental groups, which allows us to assume that the observed effects on the evaluated parameters are only due to the intentionally added AFB1, and not to the presence of another type of mycotoxins. In any case, we are definitely going to implement this evaluation in the next experiments. Thank you.

Round 2
Reviewer 1 Report
The current manuscript can be accepted in present form. No addition questions from my end.